# Booster-SHOT: Boosting Stacked Homography Transformations for Multiview Pedestrian Detection with Attention

## Abstract

Improving multi-view aggregation is integral for multi-view pedestrian detection, which aims to obtain a bird's-eye-view pedestrian occupancy map from images captured through a set of calibrated cameras. Inspired by the success of attention modules for deep neural networks, we first propose a Homography Attention Module (HAM) which is shown to boost the performance of existing end-to-end multiview detection approaches by utilizing a novel channel gate and spatial gate. Additionally, we propose Booster-SHOT, an end-to-end convolutional approach to multiview pedestrian detection incorporating our proposed HAM as well as elements from previous approaches such as view-coherent augmentation or stacked homography transformations. Booster-SHOT achieves 92.9% and 94.2% for MODA on Wildtrack and MultiviewX respectively, outperforming the state-of-the-art by 1.4% on Wildtrack and 0.5% on MultiviewX, achieving state-of-the-art performance overall for standard evaluation metrics used in multi-view pedestrian detection. [1]

## 1 Introduction

Multi-view detection Sankaranarayanan et al. (2008); Aghajan & Cavallaro (2009); Hou et al. (2020b) leverages multiple camera views for object detection using synchronized input images captured from varying view angles. Compared to a single-camera setup, the multi-view setup alleviates the occlusion issue, one of the fundamental problems in many computer vision applications. In this work, we consider the problem of multi-view pedestrian detection. As shown in Figure 1, a bird's-eye-view representation is obtained with the synchronized images from multiple calibrated cameras, which is then further used to detect pedestrians in the scene.

A central problem in multi-view detection is to obtain a correct *multi-view aggregation*. The change in viewpoint and occlusions make it challenging to match object features across different view angles. Various works attempted to address this problem, ranging from early approaches leveraging "classical" computer vision Alahi et al. (2011), hybrid approaches further incorporating deep learning, to end-to-end trainable deep learning architectures Hou et al. (2020b); Hou & Zheng (2021); Song et al. (2021).

One core challenge in multiview detection is designing how the multiple views should be aggregated. MVDet Hou et al. (2020b) proposes a fully convolutional end-to-end trainable solution for the multi-view detection task. MVDet aggregates different views by projecting the convolution feature map via perspective transformation to a single ground plane and concatenating the multiple projected feature maps. Given the aggregated representation, MVDet applies convolutional layers to detect pedestrians in the scene. Song *et al.* Song et al. (2021) identified that the projection of the different camera views to a single ground plane is not accurate due to misalignments. Consequently, they proposed to project the feature maps onto different height levels according to different semantic parts of pedestrians. Additionally, they use a neural-network-based soft-selection module to assign a likelihood to each pixel of the features extracted from the different views. They termed their approach SHOT, due to the use of the Stacked HOmography Transformations. MVDeTr Hou & Zheng (2021) extends MVDet by introducing a shadow transformer to attend differently at different positions to deal with various shadow-like distortions as well as a view-coherent data augmentation

---

[1]Code will be made public after publication and is available in the supplementary.

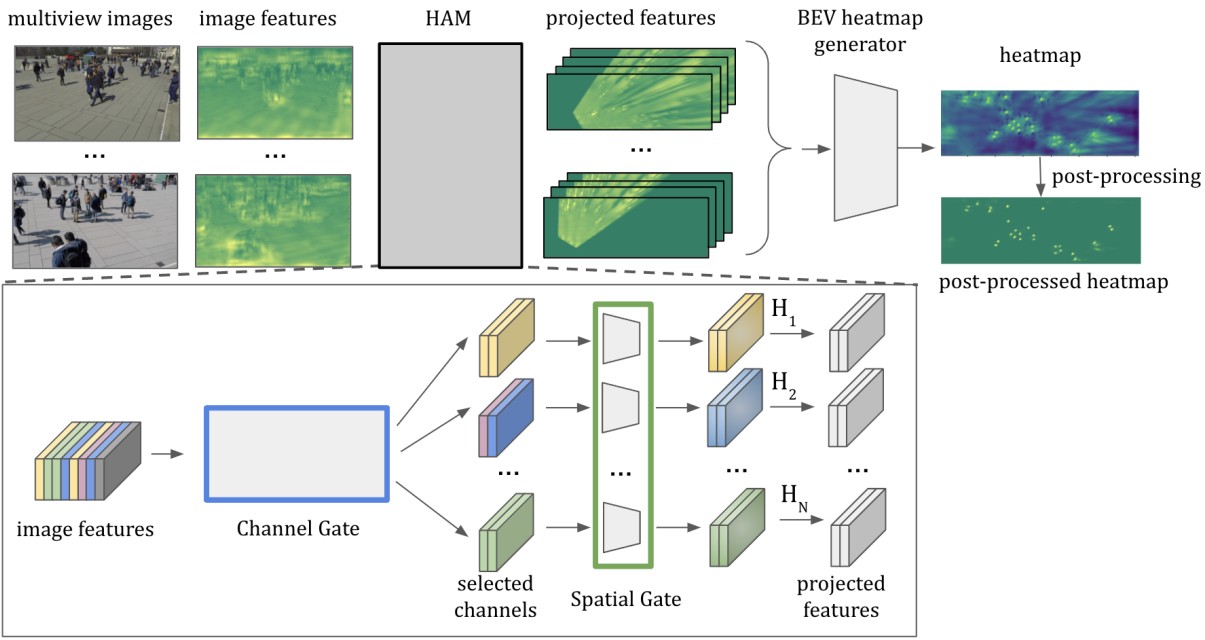

Figure 1: Overview of multiview detection with homography attention module (HAM)

method, which applies random augmentations while maintaining multiview-consistency. MVDeTr currently constitutes the SotA approach for multiview detection.

In recent years the attention mechanism for deep neural networks has played a crucial role in deep learning Hu et al. (2018); Woo et al. (2018); Guo et al. (2021b) due to the non-trivial performance gains that it enabled. Attention mechanisms have provided benefits for various vision tasks, e.g. image classification Hu et al. (2018); Woo et al. (2018), object detection Dai et al. (2017); Carion et al. (2020), semantic segmentation Fu et al. (2019); Yuan et al. (2021), or Point Cloud Processing Xie et al. (2018); Guo et al. (2021a). However, to this date, no dedicated attention mechanism has been proposed for the task of multiview pedestrian detection.

In this work, we fill this gap and propose an attention mechanism specifically designed to boost existing multiview detection frameworks. Our proposed Homography Attention Module (HAM) is specifically tailored for the core task of multiview aggregation in modern multiview detection frameworks. As shown in the lower part of Figure 1 our proposed solution consists of a channel gate module and a spatial gate module. The channel gate is directly applied to the accumulated image features from the different views. The intuition behind our *channel gate* is that different channels hold meaningful information for different homographies. The channel gate is followed by our *spatial gate*. We conjecture, that for each view and homography combination different spatial features are of higher importance. Our proposed attention mechanism can be readily plugged into existing methods.

We also combine insight from previous approaches and HAM to propose Booster-SHOT, a new end-to-end multiview pedestrian detection framework. Our experimental results show that both incorporating HAM into previous frameworks and Booster-SHOT improves over previous multiview detection frameworks and achieves state-of-the-art performance. Additionally, we provide quantitative and qualitative results to verify and justify our design choices.

## 2 Related work

### 2.1 Multiview Detection

It can be difficult to detect pedestrians from a single camera view, due to crowded scenes and occlusions. Hence, many research works study pedestrian detection in the multi-camera setup. Multiple calibrated and synchronized cameras capturing a scene from different view angles can provide a richer representation of the environment. Camera calibrations additionally produces a correspondence between each location on the ground plane and its bounding boxes in multiple camera views. This is possible since a 2D bounding box can be calculated, once an average human height and width is assumed via perspective transformation.

Research in multiview pedestrian detection has been explored intensively in the past. Early methods mainly relied on background subtraction, geometric constraints, occlusion reasoning, *etc.* Fleuret et al. (2007); Sankaranarayanan et al. (2008); Berclaz et al. (2011). Given different views from two to four video streams Fleuret *et al.* Fleuret et al. (2007), first estimates a probabilistic occupancy map of the ground plane via a generative model, which is followed by a tracking mechanism to track up to six individuals. The work by Sankaranarayanan *et al.* Sankaranarayanan et al. (2008) emphasizes how geometric constraints in multi-camera problems can be leveraged for detection, tracking, and recognition. Similar to Fleuret et al. (2007), Coates and Ng Coates & Ng (2010) also leverage a probabilistic method to fuse the outputs of multiple object detectors from different views to enhance multi-view detection in the context of robotics. Using the k-shortest paths algorithm Berclaz *et al.* Berclaz et al. (2011) propose a multiple object tracking framework, requiring as input an occupancy map from a detector. Their algorithm handles unknown numbers of objects while filtering out false positives and bridging gaps due to false negatives. Similar to previous approaches, given the output of several object detectors for different viewpoints Roig *et al.* Roig et al. (2011) estimate the ground plane location of the objects. They model the problem via Conditional Random Fields (CRFs) to simultaneously predict the labeling of the entire scene.

With the success of deep learning, deep neural networks have also been successfully applied to the multi-view detection problem Chavdarova & Fleuret (2017); Baqué et al. (2017); Hou et al. (2020b); Song et al. (2021); Hou & Zheng (2021). Chavdarova and Fleuret Chavdarova & Fleuret (2017) propose an end-to-end deep learning architecture that combines the early layers of pedestrian detectors for monocular views with a multiview network optimized for multi-view joint detection. Baqué *et al.* Baqué et al. (2017) identify that the performance of multiview systems degrades significantly in crowded scenes. To overcome this, they propose a hybrid approach of CRFs and CNNs. To perform robustly in crowded scenes their end-to-end trainable model leverages high-order CRF to model occlusions.

Recently, MVDet Hou et al. (2020b), an end-to-end trainable multiview detector has been proposed. To aggregate cues from multiple views, MVDet transforms feature maps obtained from multiple views to a single ground plane. To aggregate information from spatially neighboring locations, MVDet leverages large kernel convolutions on the multiview aggregated feature map. MVDet has been further extended through stacked homographies and shadow transformers Song et al. (2021); Hou & Zheng (2021) Song *et al.* Song et al. (2021) identified that the projection to a single view in MVDet Hou et al. (2020b) leads to inaccuracies in the alignments. Consequently, they propose to approximate projections in 3D world coordinates via a stack of homographies. Additionally, to assemble occupancy information across views, they propose a soft selection module. The soft-selection module predicts a likelihood map that assigns each pixel of the extracted features extracted from individual views to one of the homographies. MVDeTr Hou & Zheng (2021), adopts shadow transformers to aggregate multiview information, which attend differently based on position and camera differences to deal with shadow-like distortions. Additionally, MVDeTr introduces a view-coherent data augmentation method, which applies random augmentations while maintaining multiview consistency. To the best of our knowledge, MVDeTr currently constitutes the SOTA approach for multiview pedestrian detection.

### 2.2 Attention Mechanism in Computer Vision

Attention mechanisms for computer vision emphasize more important regions of an image or a feature map and suppress less relevant parts Guo et al. (2021b). They can be broadly divided into channel attention,

spatial attention, and a combination of the two variants.

**Channel Attention** selects important channels through an attention mask across the channel domain. Pioneered by Hu *et al.* Hu et al. (2018) various works have extended upon the Squeeze-and-Excitation (SE) mechanism module Gao et al. (2019); Lee et al. (2019); Yang et al. (2020); Qin et al. (2021) .

**Spatial Attention** selects important spatial regions of an image or a feature map. Early spatial attention variants are based on recurrent neural networks (RNN) Mnih et al. (2014); Ba et al. (2014). In the literature various variants of visual attention-based model can be found Xu et al. (2015); Oktay et al. (2018) To achieve transformation invariance while letting CNNs focus on important regions, Spatial Transformer Networks Jaderberg et al. (2015) had been introduced. Similar mechanisms have been introduced in deformable convolutions Dai et al. (2017); Zhu et al. (2019). Originating from the field of natural language processing, self-attention mechanisms have been examined for computer vision applications Wang et al. (2018); Carion et al. (2020); Dosovitskiy et al. (2020); Chen et al. (2020); Zhu et al. (2020).

**Channel Attention & Spatial Attention** can also be used in combination. Residual Attention Networks Wang et al. (2017) extend ResNet He et al. (2016) through a channel & spatial attention mechanism on the feature representations. A spatial and channel-wise attention mechanism for image captioning has been introduced in Chen et al. (2017). The Bottleneck Attention Module (BAM) Park et al. (2018) and Convolutional Block Attention Module (CBAM) Woo et al. (2018) both infer attention maps along the channel and spatial pathway. While in the previous two methods the channel and spatial pathways are computed separately, triplet attention Misra et al. (2021) was introduced to account for cross-dimension interaction between the spatial dimensions and channel dimension of the input. Channel & spatial attention has also been applied in the context of segmentation Roy et al. (2018); Fu et al. (2019) Further combinations of channel and spatial attention include self-calibrated convolutions Liu et al. (2020), coordinate attention Hou et al. (2021) and strip pooling Hou et al. (2020a).

## 3 Preliminaries

Let the input images for N camera views be $(I^1, \dots, I^N)$. The respective feature maps obtained from the feature extractor in the initial step of the general framework are denoted as $(F^1, \dots, F^N)$. The intrinsic, extrinsic parameters of the $i$'th camera are $\mathbf{G}^i \in \mathbb{R}^{3\times 3}$ and $\mathbf{E}^i = [\mathbf{R}^i|\mathbf{t}^i] \in \mathbb{R}^{3\times 4}$, respectively, where $\mathbf{R}^i$ is the $3 \times 3$ matrix for rotation in the 3D space and $\mathbf{t}^i$ is the $3 \times 1$ vector representing translation. Following MVDet Hou et al. (2020b), we quantize the ground plane into grids and define an additional matrix $\mathbf{F}^i \in \mathbb{R}^{3\times 3}$ that maps world coordinates to the aforementioned grid. While the mathematical concept of homography is an isomorphism of projective spaces, we use the term homography to describe correspondence relations between points on a given plane parallel to the ground as they are seen from the bird's-eye-view and from a separate camera-view. This is in line with SHOT Song et al. (2021) where the authors explain their projections as being homographies describing the translation of a plane for the pin-hole camera model. We will go into further depth regarding the homography transforms in our supplementary materials.

## 4 Methodology

### 4.1 Previous Multiview Detection Methods

Before presenting our proposed attention module, we outline the previous multiview detection frameworks in which we have implemented and tested its performance. MVDet Hou et al. (2020b) presented a multiview detection framework that functions as follows: First, the input images from different viewpoints are passed through a generic feature extractor such as ResNet18 with minor modifications. The feature maps are passed through an additional convolutional neural network that detects the head and feet of pedestrians before to aid the network during training. Next, the feature maps are projected to the ground plane via homography transformation and concatenated. Additionally, $x, y$ coordinate maps are concatenated to the stack of transformed feature maps as in CoordConv Liu et al. (2018). Finally, this is passed through a CNN to output a bird's-eye-view (BEV) heatmap which is then post-processed via thresholding and non-maximum suppression. Extending upon MVDet, MVDeTr Hou & Zheng (2021) proposed the use of affine transformations (rotation, translation, sheer, scale, cropping), which are view-coherent augmentations.

Table 1: Settings for each approach

| Method | Aug. | Loss | BEV gen. | Multi Homogr. |
|---|---|---|---|---|
| MVDet Hou et al. (2020b) | ✗ | MSE | CNN | ✗ |
| SHOT Song et al. (2021) | ✗ | MSE | CNN | ✓ |
| MVDeTr Hou & Zheng (2021) | ✓ | Focal | Transformer | ✗ |
| Booster-Shot | ✓ | Focal | CNN, Transformer | ✓ |

Additionally, the final CNN to generate the BEV heatmap is replaced with a shadow transformer, with the purpose to handle various distortion patterns during multiview aggregation. MVDeTr further replaces the MSE loss used in MVDet with Focal Loss Law & Deng (2018) coupled with an offset regression loss. While MVDet and MVDeTr both project the feature maps to the ground plane, SHOT Song et al. (2021) proposes to approximate projections in 3D world coordinates via a stack of homographies. In line with MVDet, SHOT uses a ResNet18 as a feature extractor. Contrary to MVDet, SHOT introduces additional planes parallel to the ground plane with different distances to the ground. The features are selectively projected from the camera-view to these different bird's-eye-views. As the projection to some planes may be of more importance than others, SHOT introduces a soft selection module where a network learns which homography should be used for which pixel.

The soft selection module has two shortcomings when compared to HAM. First, it uses softmax activation, therefore each pixel gets projected to some extent to each homography. Since even the homography given the lowest score by the soft selection module affects the projected outcome, this introduces some noise into the final projected feature map. In addition, all feature channels corresponding to a single pixel are multiplied by the same value when projected to a homography. However, different channels attend to different features and some will be useful for the selected homography while others won't. In contrast, HAM selects channels in a discrete manner for each homography to avoid the two problems mentioned above.

To show the efficiency of our homography attention module (HAM), we use the approaches as is without modification to their loss or training configuration, and simply plug in our proposed HAM. As the soft selection module in SHOT is rendered obsolete by our proposed HAM, we remove it when comparing the performance of SHOT with our module with the reported values for SHOT. Additionally, based on the advances made in previous works and our proposed attention module we further propose Booster-Shot (see Section 4.3).

### 4.2 Homography Attention Module

In this section, we introduce our proposed homography attention module (HAM) for boosting the performance of multi-view pedestrian detection. HAM consists of a channel gate and several spatial gates equal to the number of homographies used. Note, that our attention module is specifically designed for view-aggregation in the context of multiview detection and is hence only applied in the multiview aggregation part. The image feature maps are first passed through the channel gate, then the spatial gate, and then finally through the homography, followed by the BEV heatmap generator.

**Channel Gate** Our proposed channel gate follows the intuition that depending on the homography, different channels are of importance. Taking into consideration the multiple homography layers deployed at different heights, different feature information becomes more valuable. For instance, when we consider the homography at $Z = 0$, discriminative feature information near the ground plane, such as a person's feet, ankles, and lower legs, may offer more significant representation. This is because the homography at $Z = 0$ focuses on objects that are closer to the ground plane, which makes features near the ground plane more informative. This is in contrast to the approach proposed by SHOT, which feeds all feature maps through each of the homographies. Figure 2 outlines the architecture of our proposed channel gate, which broadly consists of the *channel selection module* and the *top-K selection module*. Given the stack of feature maps acquired from the different views, first the channel selection module is applied. The channel selection module first

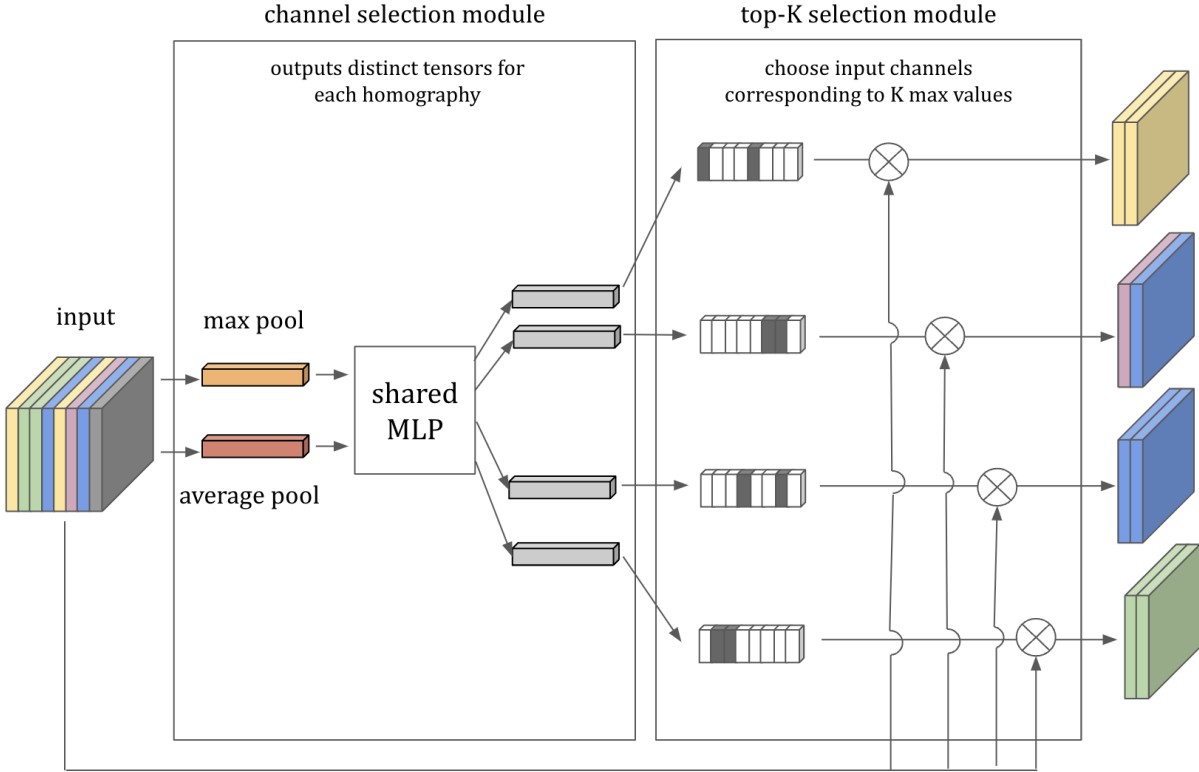

Figure 2: Diagram showing our proposed channel gate.

applies max pooling and average pooling along the spatial dimension. Both pooled feature maps are passed through a shared 2-layer MLP. As the number of channels in the output from the last layer of the MLP is decided by the number of homographies (denoted as $D$) with the number of channels in the input (denoted as $C$), we obtain $C$ channel for each homography, or in other words a $C \times D$ channel output size. Afterward, we apply the softmax function along the channel dimension for each of the outputs. The outputs are then fed into the top-K selection module. The top-K selection module takes these $D$ different $C$-dimensional outputs and selects the top $K$ largest values. The corresponding top-$K$ selected channels from the original input are then concatenated, resulting in a subset of the original input with $K$ channels. In the case of $D = 1$ (using only one homography, usually the ground plane), the top-K selection module defaults to an identity function. To retain the channel-wise aspect of our module, in this scenario we multiply the output of the channel selection module element-wise with the input. This completes the channel gate, which outputs $D$ times $K$-channel feature maps, which are then fed into spatial gate.

**Spatial Gate**  Our spatial gate is motivated by our conjecture that for each view and homography combination different spatial features are of different importance. This intuition is based on the understanding that the view determines the camera's position and orientation, the homography corresponds to different heights in the scene, and the spatial features capture the patterns, textures, and shapes of the objects in the scene. Depending on the specific view and homography combination, certain spatial features may be more informative and relevant for feature extraction than others. For example, features closer to the lower image border might be more important for a view-homography combination with a nearly parallel to the ground plane and the homography at $Z = 0$. By using a spatial gate to selectively weight and filter the spatial features for each combination, our proposed method can effectively capture the relevant information from the image and improve performance. Figure 3 shows the architecture of our spatial gate. The input is max and average pooled along the channel dimension, then concatenated channel-wise. This 2-channel

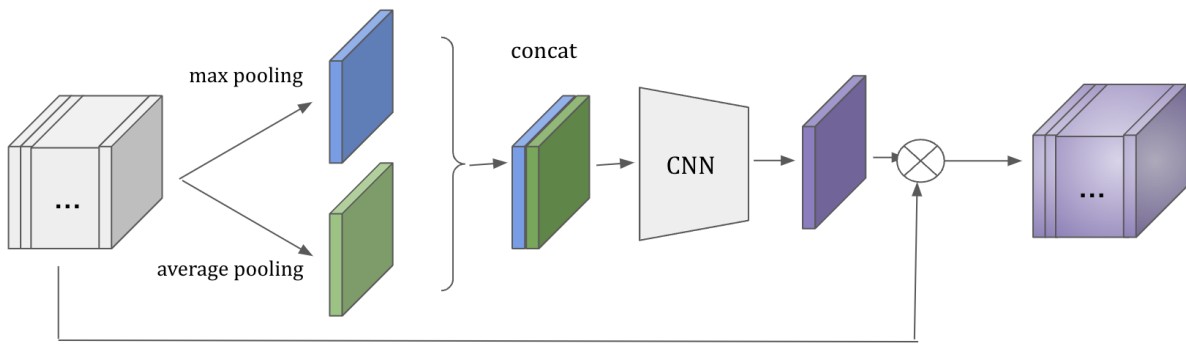

Figure 3: Diagram showing our proposed spatial gate.

input is then passed through a 2-layer convolutional neural network to generate the spatial attention map. Finally, this spatial attention map is multiplied with the original input element-wise to create an output with dimensions identical to the input. For each homography-path a separate spatial gate is applied.

Architecture-wise, while SHOT uses a "soft selection module" to estimate the importance of each homography plane for each pixel, HAM estimates the importance of channels and spatial information for each homography. Also, while MVDeTr introduced a "shadow transformer" after the homography transforms to remove shadow-like background noise, HAM uses attention to optimize image features fed to each homography and is applied prior to the homography transforms.

### 4.3 Booster-SHOT

Given the insights collected from previous approaches in addition to our proposed HAM, we design a multiview pedestrian detection architecture, which we term Booster-SHOT. Booster-SHOT bases itself on SHOT Song et al. (2021), using their stacked homography approach and leverages MVDeTr's Focal loss and offset regression loss along with the view-coherent augmentation. We retain SHOT's convolutional architecture used to generate the BEV heatmap but remove the soft selection module as the implementation of our module renders it obsolete. Figure 1 outlines how our proposed module is implemented in Booster-Shot. Table 1 outlines the design choices of Booster-SHOT alongside previous methods.

## 5 Experiments

### 5.1 Datasets

Our method is tested on two datasets for multiview pedestrian detection.
**Wildtrack** Chavdarova et al. (2018) consists of 400 synchronized image pairs from 7 cameras, constituting a total of 2,800 images. The images cover a region with dimensions 12 meters by 36 meters. The ground plane is denoted using a grid of dimensions $480 \times 1440$, such that each grid cell is a 2.5-centimeter by 2.5-centimeter square. Annotations are provided at 2fps and there are, on average, 20 people per frame. Each location within the scene is covered by an average of 3.74 cameras.
**MultiviewX** Hou et al. (2020b) is a synthetic dataset created using human models from PersonX Sun & Zheng (2019) and the Unity engine. It consists of $1080 \times 1920$ images taken from 6 cameras that cover a 16-meter by 25-meter area. Per the method adopted in Wildtrack, the ground plane is represented as a $640 \times 1000$ grid of 2.5-centimeter squares. Annotations are provided for 400 frames at 2fps. An average of 4.41 cameras cover each location, while an average of 40 people are present in a single frame.

## 5.2 Settings and metrics

In accordance with the previous methods, we report the four metrics: Multiple Object Detection Accuracy (MODA), Multiple Object Detection Precision (MODP), precision, and recall. Let us define $N$ as the number of ground truth pedestrians. If the true positives (TP), false positives (FP) and false negatives (FN) are known, precision and recall can be calculated as $\frac{TP}{FP+TP}$ and $\frac{TP}{N}$, respectively. MODA is an accuracy metric for object detection tasks and is therefore obtained by calculating $1 - \frac{FP+FN}{N}$. MODP is computed with the formula $\frac{\sum 1-d[d<t]/t}{TP}$ where $d$ is the distance from a detection to its ground truth (GT) and $t$ is the threshold for a correct detection. We keep the original threshold of 20 that was proposed in SHOT. Our implementation is based on the released code for MVDet Hou et al. (2020b), SHOT Song et al. (2021), MVDeTr Hou & Zheng (2021) and follows the training settings (optimizer, learning rate, *etc.*) for each. For all instances, the input images are resized to $720 \times 1280$ images. The output features are $270 \times 480$ images for MVDet and SHOT and $90 \times 160$ for MVDeTr. $\Delta z$ (the distance between homographies) is set to 10 on Wildtrack and 0.1 on MultiviewX. All experiments are run on two A30 GPUs (depending on the framework) with a batch size of 1.

For experiments implementing our module in SHOT, our base approach involves selecting the top-32 channels each for 4 homographies. We note that SHOT's base approach uses 5 homographies.

Table 2: Performance comparison (in %) on Wildtrack and MultiviewX datasets

| Method | Wildtrack | | | | MultiviewX | | | |
|---|---|---|---|---|---|---|---|---|
| | MODA | MODP | precision | recall | MODA | MODP | precision | recall |
| MVDet | 88.2 | 75.7 | 94.7 | 93.6 | 83.9 | 79.6 | 96.8 | 86.7 |
| MVDet + HAM | **89.6 ± 0.35** | **80.4 ± 0.21** | **95.7 ± 1.06** | **93.8 ± 0.42** | **91.3 ± 0.35** | **81.7 ± 0.14** | **98.3 ± 0.49** | **91.9 ± 2.26** |
| SHOT | 90.2 | 76.5 | 96.1 | 94.0 | 88.3 | 82.0 | 96.6 | 91.5 |
| SHOT + HAM | **90.2 ± 0.49** | **77.4 ± 0.57** | **96.2 ± 0.07** | 93.9 ± 0.42 | **91.2 ± 0.53** | **86.9 ± 4.14** | **98.2 ± 1.25** | **92.9 ± 0.78** |
| MVDeTr | 91.5 | 82.1 | 97.4 | 94.0 | 93.7 | 91.3 | **99.5** | 94.2 |
| MVDeTr + HAM | **92.8 ± 0.49** | **82.4 ± 0.71** | 96.6 ± 0.85 | **96.6 ± 1.25** | **94.2 ± 0.07** | **91.4 ± 0.57** | 99.4 ± 0.21 | **94.8 ± 0.21** |
| Booster-Shot + Tr | **92.5 ± 0.64** | 82.0 ± 0.71 | 96.3 ± 0.78 | **96.3 ± 1.50** | 93.8 ± 0.49 | 91.8 ± 0.07 | 98.8 ± 0.64 | **95.0 ± 1.06** |
| Booster-Shot | 92.8 ± 0.17 | **84.9 ± 4.42** | **97.5 ± 1.25** | 95.3 ± 1.17 | **94.4 ± 0.18** | **92.0 ± 0.04** | **99.4 ± 0.07** | 94.9 ± 0.21 |

## 5.3 Comparison with previous methods

As shown above in Table 2, we provide a comparison for the three most recent methods before and after applying our module. Applying our module to MVDet, SHOT and MVDeTr improved (or matched) all four metrics reported in their respective papers for MultiviewX. Specifically, the average performance of MVDet with our module improves over the reported values for MVDet on MultiviewX by 7.4%, 2.1%, 1.5%, and 5.2% for MODA, MODP, precision, and recall respectively. For Wildtrack, the use of our module again improved all four metrics with the exception of MVDeTr. For MVDeTr, our precision was still comparable with the reported value as there was only a 0.8% decrease in precision while the MODA, MODP, and recall each improved 1.3%, 0.3%, and 2.6% respectively. When compared with MVDet, SHOT and MVDeTr, Booster-SHOT outperforms them on all metrics except for precision against MVDeTr.

As MVDeTr proposed the shadow transformer as a way to improve performance, we applied it to Booster-SHOT and the results are denoted in Table 2 as Booster-SHOT + Tr. However, we were unable to obtain any meaningful improvement over the purely convolutional approach.

## 5.4 Ablation Experiments

**Number of homographies**  As shown in SHOT Song et al. (2021), as using multiple homographies is essentially a quantized version of a 3D projection, using more homographies leads to better performance for multi-view pedestrian detection. As our method assigns fewer channels to each homography as the number of homographies increases, we test the performance of SHOT with our module implemented for 2, 4, 6, and 8 homographies. Overall, all four metrics show improvement as the number of homographies increases

(see Table 3). The 6 homography case has the highest MODP and recall while the 8 homography case has the highest precision. Both cases mentioned above have the highest MODA. As the overall performance is very similar, we conclude that the improvement from the increased number of homographies has reached an equilibrium with the decreased number of channels passed to each homography.

Table 3: Performance depending on the number of homographies

| Method | #H | MultiviewX | | | |
| --- | --- | --- | --- | --- | --- |
| | | MODA | MODP | precision | recall |
| SHOT | 5 | 88.3 | 82.0 | 96.6 | 91.5 |
| SHOT + HAM | 2 | 89.4 | 80.8 | 95.2 | 94.2 |
| SHOT + HAM | 4 | 90.6 | 82.2 | 96.8 | 93.8 |
| SHOT + HAM | 6 | **91.4** | **83.1** | 97.4 | **93.9** |
| SHOT + HAM | 8 | **91.4** | 82.6 | **97.5** | 93.8 |

**Number of top-$K$ channels**  Our approach initially determined the number of channels selected per homography based on the number of homographies and the number of input channels. For example, our base approach for 128 input channels and 4 homographies involves selecting the top-32 channels for each homography. We further test the performance of our module when we fix the number of channels selected per homography (hereon denoted as $K$ in accordance with the name top-K selection) and change the number of output channels accordingly. Setting $K = 64$ for 4 homographies and 128 input channels indicates we take the top-64 channels for each homography and output $64 \times 4 = 256$ channels. Table 4 outlines the results we get for $K = 4, 8, 16, 32, 64, 128$. For MODA, MODP and precision, using the top-16 channels for each homography outperforms the other instances with considerable margins. The top-32 instance (our base approach) improves on the top-16 instance only for recall. We conclude that our channel selection approach is effective in removing irrelevant channels and concentrating relevant information into selected channels for each homography.

Table 4: Performance depending on the number of selected channels

| Method | $K$ | MultiviewX | | | |
| --- | --- | --- | --- | --- | --- |
| | | MODA | MODP | precision | recall |
| SHOT + HAM | 4 | 90.6 | 81.8 | 97.7 | 92.7 |
| SHOT + HAM | 8 | 90.4 | 82.2 | 97.9 | 92.4 |
| SHOT + HAM | 16 | **91.8** | **82.6** | **98.9** | 92.9 |
| SHOT + HAM | 32 | 90.6 | 82.2 | 96.8 | **93.8** |
| SHOT + HAM | 64 | 90.2 | 82.2 | 96.9 | 93.2 |
| SHOT + HAM | 128 | 89.2 | 81.8 | 96.0 | 93.0 |

**Attention Mechanisms**  In Table 5, we outline the effects of the channel gate and the spatial gate on MVDet, as well as their combination (HAM). It can be observed that both the channel gate and the spatial gate individually improve the performance over MVDet. However, using the channel gate and spatial gate subsequently, in other words HAM, improves in MODA and recall while retaining similar precision compared to MVDet, leading to an overall improvement in performance.

## 5.5   Analysis

**Efficacy of HAM in comparison to existing methods**  We emphasize that the novelty of HAM lies in the architectural integration of the attention mechanism for the specific purpose of multi-view aggregation, for which, to the best of our knowledge, our work is the first. Previous attention mechanisms (e.g. CBAM Woo et al. (2018), CCG Abati et al. (2020)) are applied at the convolutional blocks in the backbone network, while

Table 5: Performance of attention modules on MVDet

| Method | Wildtrack | | | |
| --- | --- | --- | --- | --- |
| | MODA | MODP | precision | recall |
| MVDet | 88.2 | 75.7 | 94.7 | 93.6 |
| MVDet + Channel Gate | 88.8 | 76.0 | 95.1 | 93.6 |
| MVDet + Spatial Gate | 88.6 | **76.6** | **95.5** | 93.0 |
| MVDet + HAM | **89.4** | 75.7 | 95.2 | **94.1** |

Table 6: BoosterSHOT performance with HAM vs pre-existing attention mechanisms

| | MODA | MODP | precision | recall |
| --- | --- | --- | --- | --- |
| Booster-SHOT w/o attention | $93.2 \pm 0.18$ | $91.2 \pm 0.07$ | $99.4 \pm 0.04$ | $93.7 \pm 0.20$ |
| Booster-SHOT (SE) | $93.7 \pm 0.23$ | $88.2 \pm 5.66$ | $98.1 \pm 1.11$ | $95.5 \pm 0.90$ |
| Booster-SHOT (CBAM) | $93.2 \pm 0.14$ | $90.5 \pm 0.14$ | $98.5 \pm 0.53$ | $94.7 \pm 0.35$ |
| Booster-SHOT (CCG) | $93.4 \pm 0.18$ | $91.4 \pm 0.04$ | $99.1 \pm 0.11$ | $94.2 \pm 0.07$ |
| Booster-SHOT | $\mathbf{94.4 \pm 0.18}$ | $\mathbf{92.0 \pm 0.04}$ | $\mathbf{99.4 \pm 0.07}$ | $\mathbf{94.9 \pm 0.21}$ |

HAM is applied after the backbone network since it is tailored toward multi-view aggregation. Consequently, HAM can be seen as complementary to existing attention mechanisms.

To illustrate the importance of the design choices of HAM we compare it with the naive integration of SENet, CBAM, and CCG into Booster-SHOT on MultiviewX. SENet, CBAM, and CCG come after the feature extractor in place of HAM. To provide a common baseline for HAM, SENet, CBAM, and CCG, we provide additional results for "BoosterSHOT without attention". This implementation is equivalent to SHOT Song et al. (2021) with Focal Loss and training-time augmentations.

As shown in Table 6, BoosterSHOT is shown to outperform all of the compared methods across the board. Only BoosterSHOT without attention shows similar results in precision, a very saturated metric for which BoosterSHOT shows only a slightly lower performance. In addition, when compared with BoosterSHOT without attention, adding CBAM, CCG, and SE showed only an increase of a maximum 0.5% in MODA, while adding HAM boosted MODA by 1.2%.

**Attention for different homographies**  We previously conjectured that the significance of each channel is different for each homography. In the following we validate this hypothesis through empirical evidence. Note that the following results are shown for the synthetic MultiviewX dataset. Since the results for the real-world Wildtrack dataset are consistent, we refer the reader to the supplementary for the Wildtrack visualizations. Figure 4 shows images created from camera view 1 of the MultiviewX dataset and containing output from the channel selection module corresponding to each homography. The channel selection module output is average pooled channel-wise (in this instance, the output for each homography contains 32 channels) and superimposed onto a grayscale version of the original image from the MultiviewX dataset. Yellow areas indicate high values in the output, indicating that the network is attending strongly to those regions. We denote the ground plane as H0 (homography 0) and number the remaining homographies accordingly. We can observe that the output from the channel selection module is homography-dependent as the yellow areas in all four images differ. We also note that the body parts with the brightest colors align with the height of the homographies. H0 highlights the feet while H1 highlights the lower body, especially around the knee area. H2 and H3 both highlight the upper body but H3 extends a bit farther upwards compared to H2. A similar phenomenon has been reported by the SHOT authors for their soft selection module. However, our channel selection module output shows more distinct highlighting of the body parts and is obtained through a completely different method. Overall, these results support the importance of selecting different channels for different homographies.

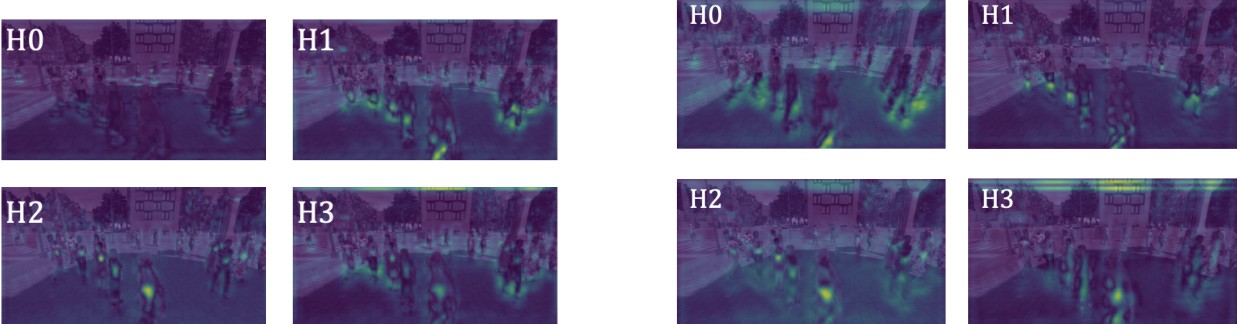

Figure 4: Homography-wise output from channel selection (left) and spatial attention maps (right)

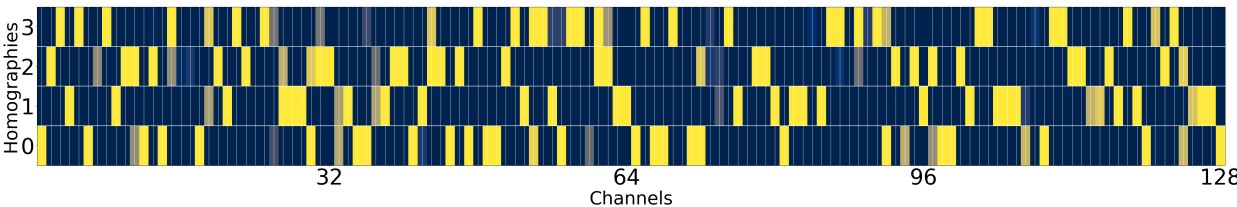

Figure 5: Heatmap representation of channel selection homography-wise. Deeper yellow colors indicates that the channel is selected most of the time while deeper blue colors are assigned to channels that are seldom selected.

Figure 4 shows the attention values from the spatial attention block at the end of our proposed module. All four attention maps show starkly different distributions, confirming our conjecture that different pixels in the feature map can differ in importance for each homography.

The results shown above were obtained through an experiment where the distance between homography planes was increased from 10cm to 60cm for MultiviewX. We noticed that, due to the low height of even the top homography plane in the 10cm case (30cm off the ground), the difference between the attention module outputs was not easily noticeable. By increasing the distance between homography planes, we were able to obtain images that clearly show homographies that are higher off the ground attend to higher regions of the human body. In addition, we noticed that the foot regression auxiliary loss caused bias toward the foot region in the extracted image features, thus distorting our heatmap visualization of the attention module outputs. As such, the experiments from which Figure 4, Figure 5 and Figure 6 were obtained did not include auxiliary losses during training (see supplementary).

We further provide results averaged over the entire MultiviewX test dataset. Specifically, we visualize how often certain channels are selected for each homography for a given view. We select Booster-SHOT for this experiment. For each channel, we count the number of times it is selected for each homography and divide by the total number of image pairs in the test set and display the resulting heatmap in Figure 5. First, it can be observed that the channels that are selected often (yellow hues) show almost no overlap across homographies, again providing evidence to our previous claim that different channels attend to different homographies. Although there are minor differences in the specific number of times some channels are chosen, the channels that are selected for the majority of the test set for each homography are unchanged (see supplementary). Interestingly, we also observe that some channels are not selected at all by any homography while other channels appear to be selected by multiple homographies.

**Attention across different views**   Figure 6 further presents evidence that our channel selection module output is only homography-dependent. We denote the camera views as C1 (camera 1) through C6 for the homography to the ground plane (H0). For all 6 images, the feet and surrounding area of the pedestrians are

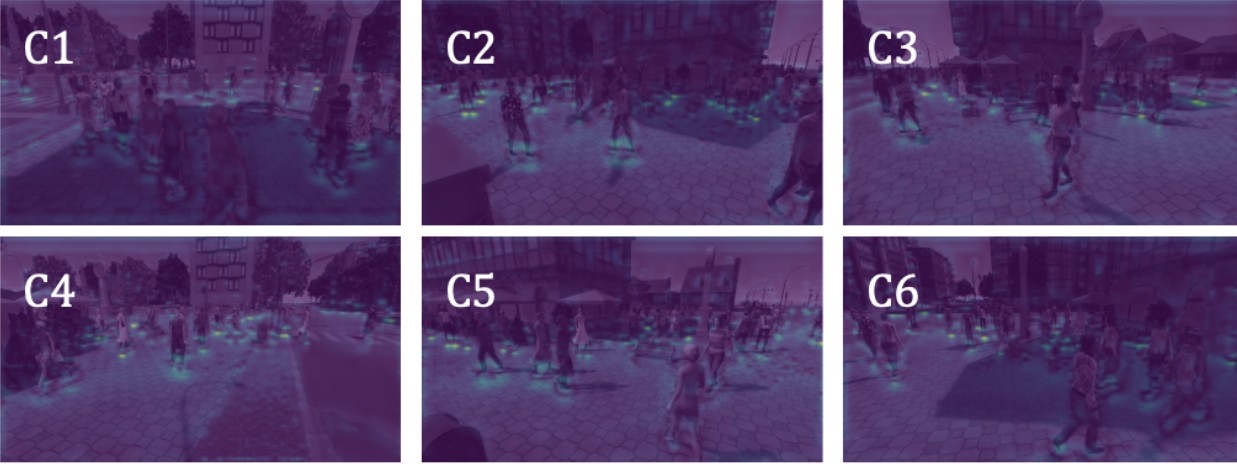

Figure 6: Camera view-wise output from channel selection module

highlighted. Therefore, we conclude that the output from the channel selection module attends consistently to certain features across all camera views.

**Generalization across camera views.** To evaluate the generalization capabilities for unseen views, we compared different methods on non-overlapping camera subsets from MultiviewX. Adding the HAM module to MVDet and SHOT showed significant improvement in MODA and precision while retaining comparable performance for MODP and recall, providing evidence that HAM facilitates generalization. For extended analysis results see supplementary.

**Computational cost, memory consumption and runtime.** We evaluate the benefits of our method in terms of computational cost, via Giga FLoating-point OPerations (GFLOPs), memory consumption, and runtime in seconds. Through one-time inference, we account for floating-point operations such as addition, multiplication, and division.[2] For evaluating memory consumption, we count model parameters and buffers. For the runtime, we ran 20 randomly generated tensors with values between 0 and 1 through the models.

With our method, upon selecting $K$ channels from the image heatmaps for each homography and using $D$ homographies, the input to the bird's-eye-view (BEV) heatmap generator has $K \times D$ channels. A smaller $K$ indicates fewer channels in the input and fewer parameters in the BEV heatmap generator. In addition, as the spatial attention module input in our spatial gate has $K$ channels, we further save on computations during channel-wise pooling.

Table 7: Computational complexity comparison between methods for 4 homographies

| Method | GFLOPs | # of parameters | runtime (seconds) |
|---|---|---|---|
| SHOT | 4.71k | 19.0M | $0.33 \pm 0.076$ |
| SHOT + HAM (top 4) | 4.09k | 14.9M | $0.29 \pm 0.071$ |
| SHOT + HAM (top 16) | 4.23k | 16.4M | $0.28 \pm 0.074$ |
| SHOT + HAM (top 32) | 4.42k | 18.5M | $0.22 \pm 0.054$ |
| MVDeTr | 2.59k | 12.8M | $0.19 \pm 0.00032$ |
| MVDeTr + HAM | 2.69k | 12.9M | $0.21 \pm 0.0029$ |
| Booster-SHOT | 2.54k | 13.3M | $0.19 \pm 0.0015$ |
| Booster-SHOT w/ Tr | 2.54k | 12.9M | $0.22 \pm 0.0016$ |

---

[2]As the post-processing cost is unchanged regardless of our settings, we take into account the calculations required to go from the RGB image input to a bird's-eye-view heatmap

Utilizing the PTFLOPS[3] package, we count the number of FLOPs for SHOT and SHOT with HAM. Table 7 shows results for SHOT and SHOT with HAM using 4, 16, and 32 channels per homography. We find that SHOT with 4 homographies, our baseline, takes up 4705.73 GFLOPs and 19048768 parameters. SHOT with HAM and 4 homographies using only the top-32 channels for each homography takes up 4423.78 GFLOPs and 18530184 parameters, yielding 6.16% improvement in computational cost and 2.63% improvement in memory usage while outperforming SHOT in all four metrics. Our best performing approach (top 16 in Table 7) takes up 4228.66 GFLOPs and 16438088 parameters, outperforming SHOT in all four metrics while achieving a 13.7% and 10.2% reduction in computational cost and memory usage. Our most lightweight approach (top 4 in Table 7) takes up 4089.63 GFLOPs and 14881112 parameters, showing a 21.6% and 13.2% reduction, respectively. It also outperforms SHOT in MODA, precision, and recall with comparable MODP (-0.2%) (see Table 4). In addition, we also tested the additional computational cost and runtime incurred by applying our HAM to previous approaches such as MVDet, SHOT, and MVDeTr. We test the computational cost and runtime of pre-existing methods before and after applying HAM. All experiments use an input tensor of size (1, 7, 3, 720, 1280) (consistent with Wildtrack/MultiviewX). ~~SHOT (4 homographies): 4705.73 GFLOPs SHOT + HAM (4 homographies): 4423.78 GFLOPs MVDeTr: 2594.53 GFLOPs MVDeTr + HAM: 2685.69 GFLOPs Booster-SHOT: 2537.40 GFLOPs~~ For SHOT, because we replaced the "soft selection module" with the lighter HAM (HAM takes fewer channels as input, reducing the number of parameters), computational cost decreases by 5.99%. We also found that by reducing the number of channels (K) selected per homography, applying HAM to SHOT can reduce the computational cost by 13.2% while still improving the performance. For MVDeTr, adding HAM (one homography) results in a 3.51% increase in computational cost. The additional cost of increasing the number of homographies is also minimal: SHOT + HAM (6 homographies): 4444.46 GFLOPs SHOT + HAM (8 homographies): 4480.92 GFLOPs. The results of the runtime evaluation overall follow the same trend as that of the computational cost.

Overall, the results indicate that HAM incurs minimal additional cost when naively applied to existing methods and enables to tune the model to reduce computational memory costs while boosting performance.

## 6  Conclusion

In this work, we propose a homography attention module (HAM) as a way to improve across all existing multiview pedestrian detection approaches. HAM consists of a novel channel gate module that selects the most important channels for each homography and a spatial gate module that applies spatial attention for each homography. In addition, we outline an end-to-end multiview pedestrian detection framework (Booster-SHOT) taking insight from previous approaches while also incorporating our proposed module. For both Booster-SHOT and previous approaches with HAM, we report new state-of-the-art performance on standard benchmarks while providing extensive empirical evidence that our conjectures and design choices are logically sound.

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
