# OpenReview forum: "Booster-SHOT: Boosting Stacked Homography Transformations for Multiview Pedestrian Detection with Attention"
_TMLR — Rejected by TMLR_

### Review · Reviewer_iVUb · 2023-02-22

**Summary Of Contributions:**

The paper proposes an attention mechanism, called the Homography Attention Module (HAM), specifically designed to improve multi-view pedestrian detection frameworks. The HAM consists of a channel gate module and a spatial gate module, and can be readily integrated into existing methods. The paper also introduces Booster-SHOT, a new end-to-end multi-view pedestrian detection framework that combines HAM and insights from previous approaches. Experimental results show that incorporating HAM into previous frameworks and using Booster-SHOT leads to improved performance and achieves state-of-the-art results. The paper fills a gap in the field of multi-view pedestrian detection, as no dedicated attention mechanism had been proposed for this task before.

**Audience:**

Yes

**Claims And Evidence:**

Yes

**Requested Changes:**

1. The paper should be more self-contained. An introduction of SHOT is helpful to understand how the proposed component is injected into existing methods.
2. Besides the theoretic FLOPs, actual running time is also needed.
3. Is is possible to integrate the proposed module into modern BEV detectors for autonomous driving? It seems the principle of multi-view aggregation is quite similar. If the authors could provide such evidence, the results could be more convincing.

**Strengths And Weaknesses:**

Pros:
1. Detailed ablation studies demonstrate the design choice of each component and parameter.
Cons:
1. The organization of the paper should be improved. For example, the paper lacks an overview of the proposed method and closely related work SHOT, which makes the readers who are unfamiliar with the context hard to follow.
2. If the top K channels are selected for each homography, how the entire method is fully end-to-end differentiable?
3. The proposed method is quite like CBAM in my view. According to the description of authors, the spatial attention module is the same as previous work, while the difference of channel attention module only lies in split the channel into homography groups. I am not sure whether these contributions meet the acceptance criteria of the journal.

---

### Review · Reviewer_BuU2 · 2023-02-23

**Summary Of Contributions:**

In this paper, the authors propose an extension to SHOT (stacked homography transformations for multiview pedestrian detection). The key difference from SHOT is that the authors designed a homography attention module (HAM) and inserted it between the multi-view feature extractor and the camera-to-BEV projection module. Albeit simple, results in Section 5 show that HAM brings consistent performance improvements to different multi-view pedestrian detection frameworks including MVDet, Shot and MVDeTr.

**Audience:**

Yes

**Broader Impact Concerns:**

The paper does not have ethical implications in my opinion.

**Claims And Evidence:**

Yes

**Requested Changes:**

The authors are expected to respond to aforementioned "weaknesses" in the revision. It is very important to distinguish HAM from existing attention modules in vision. Besides that, I'm also interested in the runtime overhead of HAM (mentioned by Reviewer iVUb) and the possibility of incorporating HAM into self-driving workloads (mentioned by the other two reviewers).

**Strengths And Weaknesses:**

## Strengths:

- The authors did a good job in summarizing previous papers in the field of multi-view detection and attention for computer vision (Sec 2, Sec 4). These sections lay a good foundation for the readers to understand the proposed method.
- The proposed method is easy to follow and it achieves consistent performance improvement as indicated in Table 2.
- The authors did spent good efforts in ablating different components in their design.

## Weaknesses:

- The proposed method has limited novelty. In the high level, I did not see too many differences between the proposed HAM and the well-known squeeze-and-excitation (SE) module. The authors also cited a bunch of attention papers in vision (p4), but they did not compare the proposed HAM with any of the method cited. This makes me question whether it is necessary to propose a new attention mechanism. One could also directly reuse existing designs such as SE, global context blocks, non-local blocks, etc.
- The authors did not seem to mention a lot of details about the backbone network used in this paper. I'm curious if the proposed HAM will still work if there are a lot of spatial/channel attention design in the backbone (e.g., ViT or SE-Nets as the backbone).
- According to the ablation studies, it seems that neither of channel gating or spatial gating is working for the recall metric. The overall improvement brought by HAM on MODP also seems to be non-existent.
- The authors did not provide error bars for all experiments. I'm not sure whether the improvement are significant enough when a fluctuation range is taken into consideration.

---

### Review · Reviewer_rHYf · 2023-02-23

**Summary Of Contributions:**

The paper proposes an attention mechanism called Homography Attention Module (HAM) designed to improve the performance of existing multiview detection frameworks for the task of multiview pedestrian detection. The proposed HAM consists of a channel gate and a spatial gate module and can be incorporated into existing methods. The authors also introduce a new end-to-end multiview pedestrian detection framework called Booster-SHOT, which combines HAM with previous approaches and achieves state-of-the-art performance. The paper provides experimental results to support the effectiveness of the proposed approach.

**Audience:**

Yes

**Claims And Evidence:**

Yes

**Requested Changes:**

Please refer to the weaknesses section for more details.

**Strengths And Weaknesses:**

---

**Strengths**:
* The proposed attention mechanism is a useful addition to existing multi-view detectors, as it can improve their performance with minimal modifications.
* The paper provides extensive experimental results on multiple benchmarks.

---

**Weaknesses**:
* The paper is not easy to read. The authors use terminology such as "homography" without adequately defining it, making it challenging for readers to understand the proposed method and its contributions. Additionally, the proposed attention module lacks proper intuition and explanation, which further complicates comprehension.
* The technical novelty of this paper is limited. The key contribution, the homography attention module, combines spatial and channel attention, both of which have been widely explored in the literature. Although the authors acknowledge this in their related work section, they should provide more information on how their proposed module differs from existing methods, as well as more experimental results to confirm its effectiveness. In my opinion, the proposed approach does not introduce any fundamentally new concepts or techniques.
* The proposed method does not seem to be limited to pedestrian detection. The authors should consider verifying the effectiveness of their proposed method on other BEV detection/segmentation tasks.

---

---

### Decision · Action_Editors · 2023-03-28

**Recommendation:** Reject

**Comment:**

Post revision, all three expert reviewers remain unconvinced about the validation of the proposed technique, as well as the scope and applications of the work.  They all recommend rejection.  The AE agrees and recommends rejection.

**Audience:**

Yes

**Claims And Evidence:**

The claims are not fully supported. Reviewers have concerns regarding whether the proposed HAM block brings advantages over many existing works.  The applications demonstrated in the submission are also quite limited.